# Association of TNF-α and IL-6 Concentrations with Depression in Patients with Rheumatoid Arthritis

**DOI:** 10.3390/cimb47060419

**Published:** 2025-06-05

**Authors:** Jelena Mrđa, Ljiljana Tadić-Latinović, Ljubinka Božić Majstorović, Vladimir Mrđa, Bosa Mirjanić-Azarić, Irma Ovčina, Semir Vranić, Snježana Popović-Pejičić

**Affiliations:** 1Department of Rheumatology, University Clinical Centre of the Republic of Srpska, 78000 Banja Luka, Bosnia and Herzegovina; buba0210@gmail.com (L.B.M.); irma.o93@hotmail.com (I.O.); 2Department of Internal Medicine, Faculty of Medicine, University of Banja Luka, 78000 Banja Luka, Bosnia and Herzegovina; spopovicpejicic@gmail.com; 3Institute of Pathology, University Clinical Centre of the Republic of Srpska, 78000 Banja Luka, Bosnia and Herzegovina; lj.tadic@yahoo.com; 4Clinic for Anesthesiology and Intensive Care, University Clinical Centre of the Republic of Srpska, 78000 Banja Luka, Bosnia and Herzegovina; mrdjavladimir@gmail.com; 5Faculty of Medicine, University of Banja Luka, 78000 Banja Luka, Bosnia and Herzegovina; bosa.mirjanic@med.unibl.org; 6Department of Clinical Biochemistry, University Clinical Centre of the Republic of Srpska, 78000 Banja Luka, Bosnia and Herzegovina; 7College of Medicine, QU Health, Qatar University, Doha P.O. Box 2713, Qatar; semir.vranic@gmail.com

**Keywords:** tumor necrosis factor α, interleukin 6, rheumatoid arthritis, depression

## Abstract

**Background/Aim:** Rheumatoid arthritis (RA) is an autoimmune inflammatory disease, characterized by the production of numerous pro-inflammatory cytokines, such as tumor necrosis factor α (TNF-α), interleukin-6 (IL-6), and interleukin-1β (IL-1β), which lead to pathophysiological changes in innate and acquired immunity. The existing evidence shows that pro-inflammatory cytokines in rheumatoid arthritis impact monoaminergic neurotransmission, neurotropic factors, and synaptic activity, which may lead to the development of depression. **Materials and Methods:** In our study, we explored the association between TNF-α and IL-6, disease activity, and the degree of depression in patients with RA. The association between TNF-α and IL-6 and the Beck and Hamilton depression scales was analyzed in a group of 116 RA patients with depression. We investigated the same correlation in 45 patients with primary depression who represented the control group. **Results:** A Spearman test showed that IL-6 levels had a positive association with the Beck and Hamilton scales (*p* < 0.05) and that TNF-α had a positive association with the Hamilton scale (*p* < 0.05). Also, the Hamilton depression scale was the more sensitive scale in the detection of depressive symptoms. **Conclusions:** Our study indicates that elevated values of pro-inflammatory cytokines are associated with the degree of depression in patients with RA. Future preclinical and clinical studies will contribute to a better understanding of the pathophysiological mechanism of depression in patients with RA and may serve as the basis for new treatment modalities. By detecting depression promptly, with the help of the HAM-D as the more sensitive scale, we could influence the future modality of treatment, and with a multidisciplinary approach, we could ensure an improvement in the quality of life of patients with RA.

## 1. Introduction

Rheumatoid arthritis (RA) occurs in about 1% of the world’s population and is one of the most common chronic autoimmune diseases, mainly affecting the joints and periarticular soft tissues [1,2]. The multi-systemic nature of RA includes the involvement of the cardiac, vascular, pulmonary, and neurological systems [3]. The pathogenesis of RA is driven by a complex interplay of pro-inflammatory cytokines, initiated through the convergence of genetic predispositions and environmental factors [4]. Autoantibodies serve as critical clinical biomarkers in systemic rheumatoid arthritis. The progression of RA is influenced by multiple other factors, including epigenetic modifications, post-translational protein alterations, aberrant glycosylation, dysregulated autophagy, and T-cell-mediated immune responses [5]. The dysregulation of innate and adaptive immunity is reflected in the production of pro-inflammatory cytokines, such as tumor necrosis factor α (TNF-α), interleukin-6 (IL-6), and interleukin-1β (IL-1β) [6,7].

Depression is one of the most common comorbidities that occur in patients with RA, and many studies show that about 50% of patients with RA have some form of depressive syndrome [8,9]. Depression in RA leads to the deterioration of functional ability, a weakened response to therapy, and an increased mortality rate [10].

Although the pathogenesis of depression has not yet been fully clarified, many studies conducted in recent years have shown immunological changes taking place in this disease [11], and many meta-analyses have shown elevated concentrations of cytokines IL-6, IL-1β, and TNF-α in the blood of patients suffering from depression in comparison to healthy controls [12,13,14,15,16]. The interaction between the peripheral immune system and the central nervous system shows convincing evidence of the role of immune-mediated inflammation in the pathogenesis of depression and RA [4].

A third of patients with depression do not respond to conventional treatment with antidepressant drugs [17], and a third of patients suffering from depression have elevated serum C-reactive protein levels (CRP > 3 mg/L) as a marker of inflammation [18]. It has been observed that the activation of the immune system, which is reflected in an increased concentration of inflammatory markers, is a predictor of a worse response to antidepressant therapy [19,20] and that inflammatory markers remain elevated over time in patients resistant to conventional antidepressant treatment [11,21]. Inflammation and the consequent activation of the tryptophan-metabolizing enzyme indoleamine 2,3-dioxygenase are thought to maintain depressive symptoms in patients with depression and elevated inflammatory markers [22]. Inflammatory cytokines, such as interferon-gamma (IFN-γ), TNF-α, and IL-6, induce indoleamine 2,3-dioxygenase [23,24]. Studies conducted on mice showed that blocking TNF-α with the drug etanercept [25] or blocking IL-6 with a monoclonal antibody [26] reduced depressive behavior after exposure to an inflammatory stimulus or stress. In addition to the humoral response, the studies also described a nervous method of interaction through the “inflammatory reflex” model, which explains that information about the peripheral immune system is transmitted to the central nervous system via the tenth cranial nerve (vagus). After that, efferent information that modifies the peripheral immune response is sent via the descending pathway [27,28].

The aim of our study was to explore the correlation between cytokines TNF-α and IL-6, disease activity, and the degree of depression in patients with RA.

## 2. Materials and Methods

The current cross-sectional study was conducted at the Internal Clinic, Rheumatology Department and Psychiatry Clinic of the University Clinical Center of the Republic of Srpska (UCC RS) from November 2021 to January 2023. It included patients aged 18 to 81 years with a confirmed diagnosis of RA who were treated at the Rheumatology Department. The diagnosis of RA was made according to the 2010 American College of Rheumatology/European League Against Rheumatism (ACR/EULAR) criteria and the ACR 1991 revised criteria [29,30]. The sample size for the study group was calculated to achieve statistical significance using G*Power software, version 3.1.9.4. We had two groups of patients: the first one was the RA patient with depression group (group I, *n* = 116), and the second was the control group (group II), which included patients with primary depression, matched in age and gender, who were treated at the Psychiatry Clinic (*n* = 45).

### 2.1. Inclusion and Exclusion Criteria

The inclusion criteria for patients with RA were as follows: (1) informed consent; (2) a diagnosis of RA confirmed by data from medical records according to criteria from the 2010 ACR/EULAR and the ACR 1991 revised criteria for functional global status in RA; [29,30] (3) moderate disease activity in patients taking a stable dose of disease-modifying antirheumatic drugs (methotrexate, leflunomide, sulfasalazine, or antimalarial) for the preceding six months (moderate disease activity is expressed by the DAS 28 index (Disease Activity Score 28) for a value of 3.2–5.1) [31]; and (4) patients who had been taking a stable dose of nonsteroidal anti-inflammatory drugs for the preceding six months. The exclusion criteria were as follows: (1) patients diagnosed with another autoimmune disease (Hashimoto thyroiditis, systemic lupus erythematosus, scleroderma, vasculitis syndromes, mixed connective tissue disease, or multiple sclerosis); (2) patients with depression diagnosed before RA; (3) patients with psychiatric disorders (schizophrenia, alcoholism, or drug addiction); (4) patients with an acute infectious disease; (5) pregnant women; (6) patients with polytrauma; and (7) patients treated with biological drugs (anti-TNF therapy or IL-6 receptor blockers).

The inclusion criteria for patients with depression were as follows: (1) informed consent; (2) the diagnosis of depression confirmed by data from medical records and mild and moderate disease forms (according to the Beck scale of depression, values of 14–19 for mild and 20–28 for moderate) [32]; and (3) patients taking a stable dose of antidepressant medication for the preceding six months. The exclusion criteria for patients with depression were as follows: (1) patients diagnosed with an autoimmune disease (RA, Hashimoto thyroiditis, systemic lupus erythematosus, scleroderma, vasculitis syndromes, mixed connective tissue disease, or multiple sclerosis); (2) patients with other psychiatric disorders (schizophrenia, alcoholism, or drug addiction); (3) patients with an acute infectious disease; (4) pregnant women; and (5) patients with polytrauma.

### 2.2. Group I Tests (RA Patients with Depressive Syndrome)

In group I, we tested the degree of disease activity based on the Disease Activity Score 28 (DAS28), clinical disease activity index (CDAI), and Simplified Disease Activity Index (SDAI), which involve a global assessment of the condition by the patient on a visual analog scale of 1 to 10 (VAS), a global assessment of the patient’s condition by a physician on a visual analog scale of 1 to 10 (VAS), and establishing the number of joints with active arthritis based on a rheumatologic examination [31,33,34]. Independently, the patients completed the Health Assessment Questionnaire (HAQ), the Rheumatoid Arthritis Quality of Life (RaQoL) questionnaire, and the 36-item Short Form Survey (SF-36) [35,36,37]. There are no defined psychiatric questionnaires to diagnose depressive syndrome in RA, so we used two scales to safely and more precisely detect depression: the Beck and Hamilton depression scales [32,38]. The Beck and Hamilton depression scales were completed by the patients with the assistance of a psychiatrist employed at the UCC RS Psychiatry Clinic.

The patients were sampled for biomarkers of inflammation—the erythrocyte sedimentation rate (ESR) and the C-reactive protein (CRP) concentration—on the same day that disease activity was assessed.

DAS 28 is an index used for assessing disease activity in patients with RA, recommended by the European Alliance of Associations of Rheumatology (EULAR) [31]. DAS 28 is calculated based on the number of tender joints (TJs) and swollen joints (SJs) out of 28 recommended joints, the patient’s assessment of their health on a visual analog scale (VAS), and the erythrocyte sedimentation rate (ESR). Recommended joints for examination are the shoulder, elbow, wrist, metacarpophalangeal, proximal interphalangeal, and knee joints. The formula for calculating the index is DAS28 = 0.56 × TJ + 0.28 × SJ + 0.7 × ln (ESR) + 0.014 × VAS, but in practice, the DAS28 calculator is used. The index result is in the range of 0.49–9.07 and interpreted as follows: disease remission (for a value of less than 2.6), low disease activity (2.6–3.2), moderate activity (3.2–5.1), and high disease activity (over 5.1) [33].

The CDAI is a clinical index of disease activity and represents a simple sum of the numbers of tender and swollen joints out of a total of 28 joints using a scale for disease assessment completed by the patient and the physician on a VAS (0–10 cm) [33]. The index value is a numeric value ranging from 0 to 76 and is interpreted as follows: disease remission (for a CDAI value less than 2.8), low disease activity (2.8–10), moderate disease activity (10–22), and high disease activity (over 22) [33].

The SDAI is a simplified index used to quickly assess disease activity in patients with RA [34]. It is calculated based on the sum of the numbers of tender and swollen joints out of a total of 28 joints, the completion of the disease assessment scale by the patient and the physician on the VAS (0–10 cm), and the CRP value. The index is a numeric value ranging from 0 to 86 and is interpreted as disease remission (for an SDAI value less than 3.3), low disease activity (3.3–11), moderate disease activity (11–26), and high disease activity (26–86) [34].

The patients’ functional abilities were examined using the Health Assessment Questionnaire (HAQ), which contains 20 questions divided into eight functional categories related to grasping, reaching for objects, dressing, personal care, getting up, feeding, usual daily activities, walking, and personal hygiene [35]. The patients filled out the questionnaire independently, answering questions about the possibility of performing certain functions as follows: 0—no difficulty; 1—with minor difficulties; 2—with major difficulties; and 3—the inability to perform the function. The total value is then divided by 20, and the HAQ index, a number with three decimal places, is calculated. The HAQ index has a three-tiered grading system: 0–1, mild impairment of function in daily life; 1–2, moderate-to-severe impairment in all segments; and 2–3, severe impairment and total disability with the need for someone else’s help. The time required to fill out the questionnaire is 10 min [35].

The Beck Depression Inventory—second edition (BDI-II) is a self-assessment measure of depressive symptoms [32]. It contains 21 items that are evaluated on a four-point scale (0–3), with higher values indicating a greater intensity of symptoms. The assessment is made concerning the preceding two weeks. The maximum number of points is 63. Results between 0 and 13 indicate minimal depression, results ranging from 14 to 19 indicate mild depression, those ranging from 20 to 28 indicate moderate depression, and results of 29 and above indicate severe depression. The psychiatrist fills out the questionnaire during the interview with the patient, with the questionnaire taking 10 min [32].

The Hamilton Depression Rating Scale (HAM-D) is one of the earliest-developed and most widely used scales that assesses the severity of depressive symptoms or the severity of a depressive episode [38]. The HAM-D has become a standard in clinical trials of depression. It is performed by a psychiatrist who interviews a patient. This scale consists of 17 items rated from 0 to 4 (0—non-existent; 4—significantly expressed). A sum of less than 7 points means that the patient is not depressed, a range of 8–13 represents mild depression, a sum of points between 14 and 18 indicates moderately severe depression, sums larger than 19 and 22 represent a severe depressive episode, and more than 23 points is a sign of very severe depression. The time required to fill out the questionnaire is 20 min [38].

The RAQoL questionnaire contains 30 questions related to the physical and psychological components of the quality of life of patients with RA [36]. Each question is scored with 1 point for a total of 30. A lower score represents a better quality of life. The patient fills out the questionnaire independently, which takes 10 min [36].

The SF 36 questionnaire (36-item Short Form Survey) is a short health status questionnaire consisting of 36 questions related to mental and physical health and social functioning, and it is frequently used for assessing quality of life [37]. The questions are divided into eight areas of examination. The advantage of this questionnaire is that it can be widely used since it does not refer to a specific age, disease, or population. The result of the test is expressed as a value in the range of 0–100, with low results showing less functionality and a worse health assessment, while high results indicate good health, without pain or functional limitations. The patient fills out the questionnaire independently; the time required to fill out the questionnaire is 20–30 min [37].

### 2.3. Group II Tests (Patients with Primary Depression)

The patients independently filled out the SF 36 quality of life questionnaire. The patients filled out the Beck and Hamilton depression scales with the help of a psychiatrist, who interpreted the results. Serum TNF-α and IL-6 were sampled from the patients on the same day.

### 2.4. Enzyme-Linked Immunosorbent Assay (ELISA)

Blood sampling was performed once during the study, before the patients completed their regular therapy, and after filling out the questionnaire. The blood was collected into a serum sample tube before immediate centrifugation at 3000× *g* for 10 min and then stored at −80 °C until required. High-sensitivity reagents, intended strictly for research, were used to establish the values of TNF-α and IL-6 in serum. In this study, we used an ELISA kit (Human IL-6 Elisa kit High Sensitivity; Human TNF Elisa kit), according to the manufacturer’s instructions. The sensitivity of the Elisa kit for TNF-α is less than 8 pg/mL. The sensitivity of the IL-6 ELISA kit is less than 0.81 pg/mL. Serum TNF-α and IL-6 levels were established at the Center for Biomedical Research of the Faculty of Medicine in Banja Luka using an ELISA device. During both ELISA tests, internal quality control measures were implemented, including the use of positive and negative control samples. The values were determined according to the tables and value ranges provided by the manufacturer. The following normal cytokine values were given: below 7.2 pg/mL for TNF-α and below 4.72 pg/mL for IL-6. The obtained results were analyzed and compared among the groups. We used the obtained parameters and values to perform the necessary analysis and statistical evaluation.

### 2.5. Ethical Approval

This study was approved by the independent Ethics Committee of the UCC RS (no 01-9-549-2/18) and the Ethics Committee of the Faculty of Medicine, University of Banja Luka (no 18/4.20/19). Before being included in the study, each patient was interviewed in detail, and then they confirmed their voluntary participation by signing and dating an informed consent form, which was also approved by the two above-mentioned ethics committees. The study was conducted following the Declaration of Helsinki.

### 2.6. Statistical Analysis

A comprehensive statistical analysis was conducted to examine clinical, demographic, and laboratory differences between two cohorts: rheumatoid arthritis (RA) patients with depression (group I) and patients with primary depression (group II). Based on normality tests (Kolmogorov–Smirnov and Shapiro–Wilk), the majority of variables in group I displayed significant deviations from normality (*p* < 0.05), necessitating the use of non-parametric methods for comparison and correlation analysis. Exceptions included the SF-36 variable, which followed a normal distribution. Descriptive statistical methods were employed in the study, along with the non-parametric tests (Mann–Whitney U test, Spearman’s rank correlation coefficient) and linear regression analysis.

These methods provided robust insights into the relationships between disease activity, inflammatory markers, and depressive symptoms in RA patients with depression versus those with primary depression.

## 3. Results

### 3.1. The Clinical, Demographic, and Laboratory Findings of the Two Cohorts

Both the Kolmogorov–Smirnov and Shapiro–Wilk normality tests yielded significant *p*-values (*p* < 0.05) across all variables in the experimental group, indicating a significant deviation from normality.

We did not identify a difference between the two examined groups concerning age and biological sex (*p* = 0.683). The female sex was more prevalent in both groups (1:4.5). The median ESR (26.50 (IQR: 15.00–38.00)) and CRP (7.00 (IQR: 2.95–12.05)) values in group I were above normal (Table 1).

The Mann–Whitney U test results showed that the distribution of functional status indicators, including DAS-28 (median: 3.80, IQR: 3.41–4.19), CDAI (median: 14.00, IQR: 11.35–16.05), and SDAI (median: 14.80, IQR: 11.75–17.33), in group I corresponded to moderate disease activity based on clinical criteria. The health status assessment questionnaire (HAQ) indicated slightly reduced physical function in group I, at 0.60 (IQR: 0.31–0.89). The median RaQoL score in group I indicated a mild-to-moderate decrease in quality of life, at 10.00 (IQR: 7.00–13.00). The Mann–Whitney U test revealed no statistically significant difference (*p* = 0.510) between groups I and II regarding the distribution of SF-36 questionnaire scores. Group I had a median score of 50.00 (IQR: 35.00–65.00), while group II had a median score of 55.00 (IQR: 30.00–80.00).

### 3.2. IL-6 Has a Strong and Positive Correlation with Disease Activity Indexes DAS28, CDAI, and SDAI

The descriptive statistics indicate that the median TNF-α value in group I was elevated at 13.88 (IQR: 11.00–16.39), whereas the median TNF-α value in group II was within the normal range at 2.41 (IQR: 1.84–2.98). The Mann–Whitney U test revealed a statistically significant difference (*p* = 0.000) in the distribution of TNF-α values between the two groups. Based on the Spearman correlation test, no statistically significant association was found between TNF-α and disease activity indexes DAS28, CDAI, and SDAI (*p* = 0.368, *p* = 0.964, and *p* = 0.911). The median value of IL-6 was elevated in group I, at 2.95 (IQR: 0.00–8.65), while it was within the normal range in group II, at 2.41 (IQR: 1.84–2.98), but no statistically significant difference was found. IL-6 had a statistically significant (*p* = 0.000), strong, and positive association with disease activity parameters DAS28, CDAI, and SDAI (Figure 1, Table 2).

### 3.3. TNF-α and IL-6 are Correlated with the Presence of Depression

In group I, IL-6 exhibited a positive and moderately strong association with the Hamilton depression scale (r = 0.433, *p* < 0.05) and the Beck depression scale (r = 0.417, *p* < 0.05). Both associations were statistically significant (*p* = 0.000) (Figure 2, Table 3).

TNF-α had a statistically significant (*p* < 0.05), positive, and weak association with the Hamilton depression scale (r = 0.196, *p* = 0.035) (Figure 3, Table 4), while no association was found with the Beck depression scale (r = 0.17, *p* = 0.068). No significant association was found between cytokines and the Hamilton and Beck depression scales in group II (*p* > 0.05).

Further statistical linear regression analysis shows that IL-6 (B = 0.171, *p* = 0.000) and TNF-α (B = 0.039, *p* = 0.012) are statistically significant predictors (*p* < 0.05) of depression intensity on the Hamilton depression scale in group I (Table 5). IL-6 had a stronger association with depression intensity. Confounders such as CRP (*p* = 0.548), gender (*p* = 0.478), and age (*p* = 0.227) had no statistically significant influence (*p* > 0.05). Residuals were checked and met the assumptions of normality (K-S *p* = 0.064 > 0.05 and S-W *p* = 0.058 > 0.05). The same analysis for the Beck depression scale showed that IL-6 (B = 0.151, *p* = 0.000) and TNF-α (B = 0.033, *p* = 0.012) were significant predictors (Table 6), while potential confounders had no statistically significant effect (*p* > 0.05).

### 3.4. Higher Disease Activity Correlated with Depression Degree in Group I

The RA activity index DAS28 had a statistically significant (*p* < 0.05) and positive but weak association with the Hamilton depression scale (r = 0.352, *p* = 0.000), while its correlation with the Beck depression scale was statistically significant, moderately strong, and positive (r = 0.378, *p* = 0.000). The clinical disease activity index, CDAI, had a statistically significant (*p* < 0.05), positive, and weak association with the Hamilton depression scale (r = 0.351, *p* = 0.000). The Beck depression scale had a statistically significant (*p* < 0.05), moderately strong, and positive association with the index of clinical disease activity (r = 0.404, *p* = 0.000). The Simplified Disease Activity Index, SDAI, had a statistically significant (*p* < 0.05), positive, and weak correlation with the Hamilton scale for depression (r = 0.357, *p* = 0.000) and a moderately strong, positive association with the Beck scale (r = 0.415, *p* = 0.000). A higher level of disease activity is associated with the degree of depression in group I.

### 3.5. Pain Is Directly Correlated with Depression

A statistically significant association between pain and the Hamilton and Beck depression scales was detected in group I (r = −0.430, *p* = 0.000; r = −0.465, *p* = 0.000) as well as between pain and the Hamilton depression scale in patients with primary depression (r = −0.309, *p* = 0039) (Figure 4, Table 7). We noticed that the group suffering from depression had a statistically significantly (*p* = 0.000) higher rank score of pain at 101.48 than the RA group at 73.06.

### 3.6. Hamilton Depression Scale as More Sensitive Scale for Detecting Depression

Regression analysis demonstrated that the HAM-D had a strong predictive relationship with BEK II scores (B = 0.738, Beta = 0.849, t = 17.173, *p* = 0.000) (Table 8). The obtained analysis led to the conclusion that twice as many patients with a moderate form of the disease were diagnosed using the Hamilton scale compared to the Beck scale in the experimental group.

### 3.7. The Duration of the Disease Does Not Impact the Severity of Depressive Symptoms

In group I, there was no significant association (*p* > 0.05) between the duration of the disease and the Hamilton (r = 0.020, *p* = 0.828) and Beck depression scales (r = 0.087, *p* = 0.351).

## 4. Discussion

In recent years, there has been growing interest in examining psychiatric diseases in patients with autoimmune diseases. Autoimmune diseases are characterized by chronic inflammation, which can cause psychiatric disorders through the inflammation of the central nervous system and neurotransmitter abnormalities. In addition to the previous findings about the influence of pain and functional disability on the onset of depression in patients with RA, there is increasing evidence in favor of a complex pathophysiological basis [1]. Lu et al. indicated a bidirectional correlation between depression and RA, where depression preceded the development of RA and vice versa. The exact mechanism is not clear, but the reason for the onset of both diseases can be found in immune system alterations and the fact that common predisposing factors can affect immune disruption [39].

Choy and Calabrese described the role of IL-6 in the development of pain and weakness in patients with RA and studied its role in patients with depression [40]. Many studies also detected a decrease in symptoms of pain, weakness, and depression if IL-6 blockers were used, but it is difficult to identify whether the effect on one symptom reduced the other symptoms as well or whether IL-6 is responsible for the occurrence of each symptom individually. Cytokines TNF-α and IL1β also impact the pathogenesis of weakness [41] and pain [42,43], and as immune mediators, they indirectly affect the pathophysiological mechanism of depressive syndrome [1]. Analyses have shown that a functional single-nucleotide polymorphism (SNP) in the promoter region of the IL-6 gene leads to elevated values of IL-6 and CRP, which correlate with the severity of depressive syndrome [44].

Patients with more severe disease had higher degrees of depression and disability, with a negative impact on quality of life. In the control group of patients with depression, we did not find a correlation between the level of pro-inflammatory cytokines and depression.

Interestingly, pain had a higher average value and correlation with depression in patients with primary depression, although pain was statistically significantly correlated with depression in both studied groups. One of the reasons could be a cytokine influence or another mechanism of change in the central nervous system, since the impact of changes in the immune system on the periphery can be significantly eliminated in patients with depression unlike the pathophysiological mechanism in RA.

Diagnosing depression in patients with RA is often difficult since many symptoms of these two diseases overlap (e.g., fatigue, weight loss, insomnia, and poor appetite), so depression remains undetected in many cases [45,46]. The current recommendations for diagnosing and monitoring RA do not offer clearly defined psychiatric questionnaires to diagnose depressive syndrome. In our research, we used two scales to identify the degree of depression, the Beck and Hamilton scales, and we discovered certain differences in the results of the studied correlations. Since the Hamilton scale detected more patients with a more pronounced form of depression than the Beck scale, we conclude that the HAM-D is more sensitive and should be used to acquire a more adequate and reliable assessment of the degree of depression. Earlier diagnosis and appropriate treatment of depression could improve patients’ health in general.

The effectiveness of biologic drugs such as cytokine antagonists in the treatment of depression could highlight the connection between inflammatory pathways and mood disorders. Some anti-TNF drugs, such as infliximab and etanercept, have been reported to decrease depressive symptoms, especially for patients with depression with higher CRP levels at baseline [47,48]. Some investigators reported rapid improvements in pain and mood levels within 24 h after anti-TNF therapy [49]. A recent systematic review and meta-analysis conducted by Baghdadi et al. suggested that therapy with an IL-6 receptor antagonist (tocilizumab, TCZ) had a beneficial impact on depression in patients with RA [50], but further research is necessary to evaluate the long-term efficacy and safety profile of TCZ in treating depression.

The unpredictable course of RA, the presence of chronic pain, and depressive syndrome affect many aspects of a patient’s life, such as social relations, family life, and work ability [51], and impair quality of life [52].

The limitations of this study could be attributed to the selection of patients with RA with a mild and moderate form of the disease and a small number of subjects. If the study included both patients with severe depressive syndrome and therapeutically naïve patients, detecting pro-inflammatory cytokines more precisely would have been possible. Furthermore, conducting the study over an extended period, incorporating multiple sequential measurements of pro-inflammatory cytokines, would facilitate the longitudinal assessment of their influence on the progression of depression.

By detecting depression promptly, with the help of the HAM-D as the more sensitive scale, we could influence the future modality of treatment, and with a multidisciplinary approach, we could ensure an improvement in the quality of life of patients with RA.

## 5. Conclusions

Our study has demonstrated a positive association between cytokines TNF-α and IL-6 and the degree of depression in patients with rheumatoid arthritis, as well as between IL-6 levels and disease activity measured by rheumatoid arthritis activity indexes. RA severity was strongly and positively correlated with depressive syndrome. Pain was positively correlated with the degree of depression in both studied groups, especially in patients with depressive syndrome. Given that TNF-α and IL-6 remain significant predictors of depression severity, their influence on depression persists even after adjusting for potential confounders such as C-reactive protein (CRP), sex, and age, thereby suggesting an independent association between these inflammatory cytokines and depression severity.

## Figures and Tables

**Figure 1 cimb-47-00419-f001:**
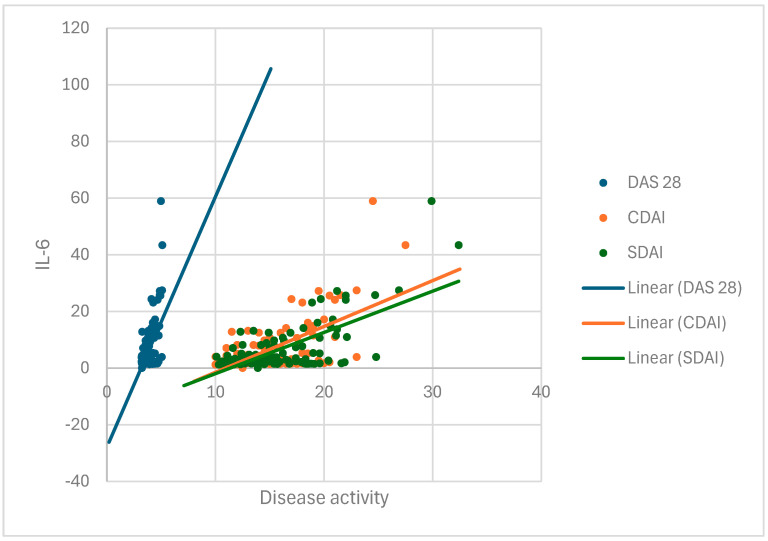
Correlation between IL-6 and disease activity in group I (RA patients with depression).

**Figure 2 cimb-47-00419-f002:**
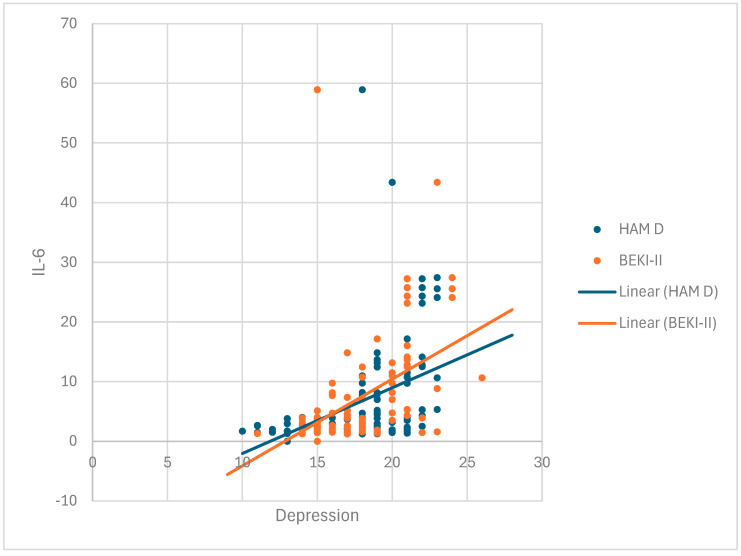
Correlation between IL-6 and depression in group I (RA patients with depression).

**Figure 3 cimb-47-00419-f003:**
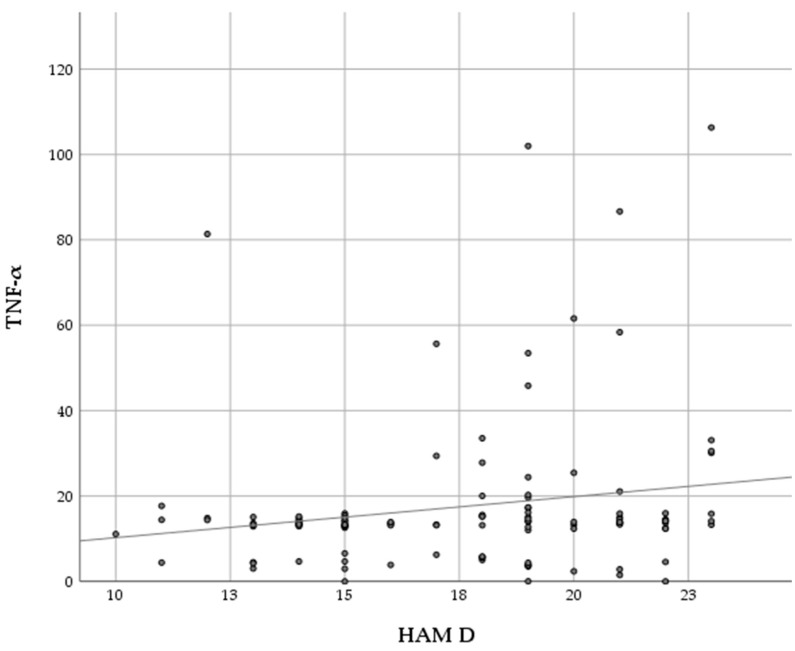
Correlation between TNF-α and depression in group I (RA patients with depression). **·** Value of TNF-α and HAM-D for RA patient with depression.

**Figure 4 cimb-47-00419-f004:**
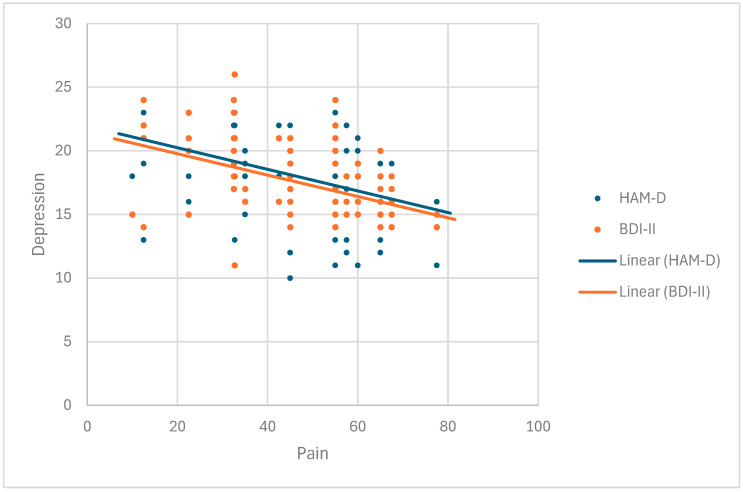
Correlation between pain and depression in group I (RA patients with depression).

**Table 1 cimb-47-00419-t001:** Demographics, clinical parameters, and test results.

Characteristic	Group I (RA Patients with Depression *n* = 116)	Group II (Patients with Primary Depression *n* = 45)	*p*-Value
**Biological sex (M/F)**	20/96	9/36	*p* = 0.683
**Age**	62.00 (IQR: 56.00–68.00)	59.00 (IQR: 52.00–66.00)	*p* = 0.900
**ESR**	26.50 (IQR: 15.00–38.00)	-	-
**CRP**	7.00 (IQR: 2.95–12.05)	-	-
**DAS28**	3.80 (IQR: 3.41–4.19)	-	-
**CDAI**	14.00 (IQR: 11.35–16.05)	-	-
**SDAI**	14.80 (IQR: 11.75–17.33)	-	-
**HAQ**	0.60 (IQR: 0.31–0.89)	-	-
**RaQoL**	10.00 (IQR: 7.00–13.00)	-	-
**HAM-D**	18.00 (IQR: 15.00–21.00)	13.00 (IQR: 9.00–17.00)	*p* = 0.00
**BDI-II**	17.00 (IQR: 15.00–20.00)	16.00 (IQR: 13.00–19.00)	*p* = 0.991
**TNF-α**	13.88 (IQR: 11.00–16.39)	3.52 (IQR: 2.16–4.87)	*p* = 0.00
**IL-6**	2.95 (IQR: 0.00–8.65)	2.41 (IQR: 1.84–2.98)	*p* = 0.075
**SF-36**	50.00 (IQR: 35.00–65.00)	55.00 (IQR: 30.00–80.00)	*p* = 0.510

The numerical data are presented as the median (25th percentile and 75th percentile). The Mann–Whitney U test was applied for comparisons between independent groups, while the Spearman correlation test was used to evaluate associations. An alpha error of 0.05 was adopted in all statistical analyses. Significant *p*-values are marked in bold. ESR—erythrocyte sedimentation rate; CRP—C-reactive protein; DAS28—Disease Activity Score 28; CDAI—clinical disease activity index; SDAI—Simplified Disease Activity Index; HAQ—Health Assessment Questionnaire; RaQoL—Rheumatoid Arthritis Quality of Life questionnaire; HAM-D—Hamilton Depression Rating Scale; BDI-II—Beck Depression Inventory—second edition; TNF-α—tumor necrosis factor α; IL-6—interleukin-6; SF-36—SF 36 questionnaire.

**Table 2 cimb-47-00419-t002:** Correlation between IL-6 and disease activity in group I (RA patients with depression).

			IL 6
Spearman’s rho	DAS28	Correlation coefficient	0.318 **
		Sig. (2-tailed)	0.000
		N	116
	CDAI	Correlation coefficient	0.397 **
		Sig. (2-tailed)	0.000
		N	116
	SDAI	Correlation coefficient	0.413 **
		Sig. (2-tailed)	0.000
		N	116

** Correlation is significant at the 0.01 level (2-tailed).

**Table 3 cimb-47-00419-t003:** Correlation between IL-6 and depression in group I (RA patients with depression).

			IL 6
Spearman’s rho	HAM D	Correlation coefficient	0.514 **
		Sig. (2-tailed)	0.000
		N	116
	BEK II	Correlation coefficient	0.534 **
		Sig. (2-tailed)	0.000
		N	116

** Correlation is significant at the 0.01 level (2-tailed).

**Table 4 cimb-47-00419-t004:** Correlation between TNF-α and depression in group I (RA patients with depression).

			HAM D
Spearman’s rho	TNF_a	Correlation coefficient	0.196 *
		Sig. (2-tailed)	0.035
		N	116

* Correlation is significant at the 0.05 level (2-tailed).

**Table 5 cimb-47-00419-t005:** IL-6 and TNF-α as predictors of depression on HAM D scale (linear regression).

Coefficients ^a^						
Model		Unstandardized Coefficients		Standardized Coefficients	t	Sig.
		B	Std. Error	Beta		
1	(Constant)	15.893	0.449		35.402	0.000
	TNF	0.039	0.015	0.213	2.562	0.012
	IL 6	0.171	0.032	0.443	5.339	0.000
2	(Constant)	15.132	2.220		6.816	0.000
	TNF	0.041	0.016	0.221	2.606	0.010
	IL 6	0.190	0.050	0.492	3.835	0.000
	CRP	−0.030	0.050	−0.080	−0.603	0.548
	Sex	−0.560	0.788	−0.063	−0.711	0.478
	Age	0.032	0.026	0.102	1.214	0.227

^a^ Dependent variable: HAM D.

**Table 6 cimb-47-00419-t006:** IL-6 and TNF-α as predictors of depression on BEK II scale (linear regression).

Coefficients ^a^						
Model		Unstandardized Coefficients		Standardized Coefficients	t	Sig.
		B	Std. Error	Beta		
1	(Constant)	15.574	0.376		41.421	0.000
	TNF_a	0.033	0.013	0.206	2.578	0.011
	IL 6	0.170	0.027	0.505	6.307	0.000
2	(Constant)	14.635	1.836		7.971	0.000
	TNF_a	0.033	0.013	0.205	2.544	0.012
	IL 6	0.151	0.041	0.450	3.683	0.000
	CRP	0.015	0.041	0.045	0.361	0.719
	Sex	−0.820	0.651	−0.107	−1.259	0.211
	Age	0.040	0.022	0.149	1.859	0.066

^a^ Dependent variable: BEK II.

**Table 7 cimb-47-00419-t007:** Correlation between pain and depression in group I (RA patients with depression).

			Pain
Spearman’s rho	HAM D	Correlation coefficient	−0.430 **
		Sig. (2-tailed)	0.000
		N	116
	BEK II	Correlation coefficient	−0.465 **
		Sig. (2-tailed)	0.000
		N	116

** Correlation is significant at the 0.01 level (2-tailed).

**Table 8 cimb-47-00419-t008:** HAM-D as scale with strong predictive relationship with BEK II (regression analysis).

Coefficients ^a^						
Model		Unstandardized Coefficients		Standardized Coefficients	t	Sig.
		B	Std. Error	Beta		
1	(Constant)	4.202	0.773		5.435	0.000
	HAM D	0.738	0.043	0.849	17.173	0.000

^a^ Dependent variable: BEK II.

## Data Availability

The data presented in this study are available upon request from the corresponding author due to privacy restrictions.

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
