# Peer review of "Association of TNF-α and IL-6 Concentrations with Depression in Patients with Rheumatoid Arthritis"

_cimb, 2025, doi:10.3390/cimb47060419_

Round 1
Reviewer 1 Report
Comments and Suggestions for Authors
Suggestions for authors:
- It would be useful to correct the title of the manuscript to „Correlation of the concentrations of TNF-α and IL-6 with depression in patients with rheumatoid arthritis“ to emphasize the primary aim of the paper
- Authors should pay attention to the abbreviation for the cytokine tumor necrosis factor alpha, the full name is missing throughout the text (state TNF-α instead TNF).
- It is desirable to interpret the applicability of research results in practice from the aspect of regression relationships of the analyzed variables, both in the summary of the manuscript and in the discussion.
- supplement the introductory part with a couple of sentences related to the pathogenesis of RA
- methodology, type of study: was it a prospective or cross-sectional study, a prospective study implies monitoring the subjects for an appropriate period of time with testing at multiple time points
- the term experimental group is rough and is mainly used as a term in animal studies, recommendation to the authors to correct the name
- reformulate the paragraph that refers to the applied statistical tests, it is enough to explain the procedures as a whole and not each applied test individually
- In the results section it is not necessary to provide data on the normality of the distribution of variables, it is sufficient to state the name of the statistical test that takes this fact into account in terms of the application of parametric or non-parametric tests
- the authors state under Table 1 that a linear regression test was performed, although this data is not shown in the table
- unify the font of the letters on figure 1 and figure 2 (for „disease activity“, „depression“..)
- present the regression results tabularly
- Limitations of the study should be stated at the end of the discussion
- did the authors investigate the effects of therapy in patients with RA on the concentration of measured cytokines
Author Response
For research article
|
Response to Reviewer 1 Comments
|
|||
|
1. Summary |
|
|
|
|
Thank you very much for taking the time to review this manuscript. Please find the detailed responses below and the corresponding revisions/corrections in red/ in the re-submitted files.
|
|||
|
|
|
||
|
|
|
||
|
|
|
||
|
|
|
||
|
|
|
||
|
2. Point-by-point response to Comments and Suggestions for Authors |
|||
|
Comments 1: It would be useful to correct the title of the manuscript to „Correlation of the concentrations of TNF-α and IL-6 with depression in patients with rheumatoid arthritis “to emphasize the primary aim of the paper.
|
|||
|
Response 1: Thank you for the suggestion — it would emphasize the primary aim of the paper. In order to comply with the strict recommendation of the second reviewer to use the term 'association' instead of 'correlation', we hope you agree with the following proposed title of the paper: 'Association of TNF-α and IL-6 concentrations with depression in patients with rheumatoid arthritis'. |
|||
|
Comments 2: Authors should pay attention to the abbreviation for the cytokine tumor necrosis factor alpha, the full name is missing throughout the text (state TNF-α instead TNF).
|
|||
|
Response 2: Thank you, we have corrected TNF to TNF-α throughout the entire text (it is marked in red).
Comments 3: It is desirable to interpret the applicability of research results in practice from the aspect of regression relationships of the analyzed variables, both in the summary of the manuscript and in the discussion.
Response 3: We sincerely appreciate your insightful suggestion; in the concluding sections of the summary and discussion, we emphasized the practical implications of our research (marked in red, line 38-41; 471-473).
Comments 4: Supplement the introductory part with a couple of sentences related to the pathogenesis of RA.
Response 4: We have added several sentences on the pathogenesis of rheumatoid arthritis in the Introduction section (line 47-55).
Comments 5: Methodology, type of study: was it a prospective or cross-sectional study, a prospective study implies monitoring the subjects for an appropriate period of time with testing at multiple time points.
Response 5: Thank you for your helpful comment. We confirm that the study is of a cross-sectional design (line 91).
Comments 6: The term experimental group is rough and is mainly used as a term in animal studies, recommendation to the authors to correct the name.
Response 6: Yes, we have changed the name of the group according to your recommendation (line 99).
Comments 7: Reformulate the paragraph that refers to the applied statistical tests, it is enough to explain the procedures as a whole and not each applied test individually
Response 7: Thank you, it is corrected (line 250-252), we have excluded the data from line 253-269.
Comments 8: In the results section it is not necessary to provide data on the normality of the distribution of variables, it is sufficient to state the name of the statistical test that takes this fact into account in terms of the application of parametric or non-parametric tests
Response 8: Thank you, it is corrected (line 275-281).
Comments 9: The authors state under Table 1 that a linear regression test was performed, although this data is not shown in the table
Response 9: We appreciate your valuable suggestion; due to an oversight, the explanation of linear regression was mistakenly included in Table 1. In the results section, however, we have presented the outcomes obtained from regression analysis in Tables 5, 6, and 8.
Comments 10: Unify the font of the letters on figure 1 and figure 2 (for „disease activity“, „depression“..)
Response 10: Thank you, it is corrected (in same font as the text in manuscript).
Comments 11: Present the regression results tabularly.
Response 11: Thank you, it is now presented in Tables 5, 6, and 8.
Comments 12: Limitations of the study should be stated at the end of the discussion.
Response 12: We appreciate your feedback and have updated the limitations of the study in accordance with your recommendation (line 464-470).
Comments 13: Did the authors investigate the effects of therapy in patients with RA on the concentration of measured cytokines
Response 13: The authors did not investigate the effect of therapy on cytokine concentrations in patients with RA. We would like to continue exploring this aspect in future studies, and we appreciate your suggestion. Targeted therapies, such as anti-TNF blockers and tocilizumab, are available for a smaller number of patients. We will monitor whether we can obtain a sufficient sample size for statistical significance.
|
|||
|
5. Additional clarifications
|
|||
|
We welcome any additional suggestions or recommendations |
|||

Reviewer 2 Report
Comments and Suggestions for Authors
Abstract requires modifications
Some Refs are missing in Introduction Section
Many Methodological Biases exist (selection biases, inclusion/exclusion criteria,etc.)
Questionnaires without Refs
Do not mantion trade marks
Chi-square model is weak and unreliable
The presentation of Results (No Tables) will confuse the median Reader
Do not use the word "correlation"... Replace by "association"
Control of possible confounders??
(The Authors must see my remarks)

Author Response
For research article
|
Response to Reviewer 2 Comments
|
||
|
1. Summary |
|
|
|
Thank you very much for taking the time to review this manuscript. Please find the detailed responses below and the corresponding revisions /corrections red/ in the re-submitted files
|
||
|
2. Point-by-point response to Comments and Suggestions for Authors |
||
|
Comments 1: Abstract requires modifications
|
||
|
Response 1: We appreciate your recommendation and have made revisions in accordance with your guidance (marked red in abstract).
|
||
|
Comments 2: Some Refs are missing in Introduction Section
|
||
|
Response 2: Agree. Thank you, we have made the revisions as per your suggestion (references are marked red in text).
Comments 3: Many Methodological Biases exist (selection biases, inclusion/exclusion criteria,etc.)
Response 3: Thank you, in the concluding section of the discussion, we have delineated and elaborated upon the limitations inherent in our study (line 464-470).
|
||
|
Comments 4: Questionnaires without Refs
Response 4: Thank you, we inadvertently included references at the end of the paragraph where we described the questionnaires in the study (references are marked red).
Comments 5: Do not mantion trade marks
Response 5: Thank you for suggestion, we have changed text in line 224-225. The ELISA testing was conducted following the manufacturer's instructions. We have provided a more detailed explanation of the ELISA test procedure to enhance clarity for the reader.
Comments 6: Chi-square model is weak and unreliable Response 6: We sincerely appreciate your insightful comment; we have repeated the testing using regression analysis (line 390-394), and the results are presented in Table 8.
Comments 7: The presentation of Results (No Tables) will confuse the median Reader
Response 7: Yes, thank you, the results are now presented in Tables 2, 3, 4, 5, 6, 7, and 8.
Comments 8: Do not use the word "correlation"... Replace by "association"
Response 8: Thank you, it is corrected in manuscript.
Comments 9: Control of possible confounders??
Response 9: We appreciate your highlighting of one of the key aspects of our study; in the conclusion, we have emphasized how TNF-α and IL-6 remain significant predictors of depression severity, their influence on depression persists even after adjusting for potential confounders such as C-reactive protein (CRP), sex, and age, thereby suggesting an independent association between these inflammatory cytokines and depression severity.
|
||
|
5. Additional clarifications
We welcome any additional suggestions or recommendations. |
||

Round 2
Reviewer 1 Report
Comments and Suggestions for Authors
I thank the authors for their detailed responses. I believe that by accepting the recent suggestions, the authors have significantly improved the quality of the manuscript.